# Dealing with discontinuous meteorological forcing in operational ocean modelling: a case study using ECMWF-IFS and GETM (v2.5)

**Bjarne Büchmann**[1]

[1]FCOO: Joint GeoMETOC Support Center, Danish Defence Acquisition and Logistics Organization (DALO), Lautrupbjerg 1-5, DK-2750 Ballerup, Denmark

**Correspondence:** Bjarne Büchmann (bjb@fcoo.dk)

**Abstract.** Meteorological data providers release updated forecasts several times per day – at the forecast epochs. The first time step ($t = 0$) of each forecast, the so-called analysis step, is updated by a data-assimilation process, so that the meteorological fields at this time in general do not match the fields from the previous forecast. Seen from the perspective of oceanographic modelling, the analysis step represents a possible discontinuity in the model forcing. Unless care is taken, this "meteorological discontinuity" may generate spurious waves in the ocean model. The problem is examined and quantified for a single meteorological model: the European Centre for Medium-range Weather Forecasts (ECMWF) Integrated Forecasting System (IFS). A simple straight-forward solution is suggested to overcome the forcing discontinuity, and the effect on two particular ocean models is examined: the FCOO NA3 (North Atlantic 3nm) storm surge model, and the NS1C (North Sea – Baltic Sea 1nm) circulation model.

## 1 Introduction

Numerical ocean models are in operational use at many institutions around the world. In the present paper, a particular issue related to an inherent discontinuity of operational meteorological forcing data will be examined. It will be shown that a naïve implementation of an operational forecast process may lead to discontinuous forcing data, which excite spurious waves in the oceanographic model. Further, unless care is taken, there could be a difference in the forcing data applied in, respectively, forecast/nowcast and hindcast modes. This difference may result in under-prediction of forecast/nowcast errors of ocean models.

The discontinuity problem is examined, and a straight-forward solution is proposed. The effects of the discontinuity on a particular ocean model are quantified, and results with and without use of the outlined solution are examined.

## 2 Background

### 2.1 Operational forecast in meteorology

Since the early days of numerical weather prediction (see *e.g.* Richardson, 1922), observations have played a major role: to accurately forecast the weather systems, an understanding of the near past is necessary. Operational weather forecasts run several times per day, using data assimilation schemes to update the initial field in a so-called "analysis step", see *e.g.* Lynch (2008). The specific methods used during the analysis is beyond the scope of the present work, but see *e.g.* Buizza et al. (2005) for a comparison between several weather centres. As we shall see, the data provided for the downstream users are significantly impacted by the assimilation and the analysis step, so ocean modellers should be aware of the consequences and take appropriate actions to deal with possible adverse effects.

As part of the optional boundaries programme, the European Centre for Medium-range Weather Forecasts (ECMWF) provides forecasts every six hours, at times which in the following will be denoted the "epoch" (time zero) of each forecast. Each forecast consists of a number of hourly "fields" (time steps), delivered in GRIB format to the member countries. For each forecast, the simulated time may be denoted by the number of hours from the epoch. Thus, +6H corresponds to the time of the following epoch (the base time of the next/coming forecast), while -6H is the previous epoch. As each forecast is significantly longer than six hours,

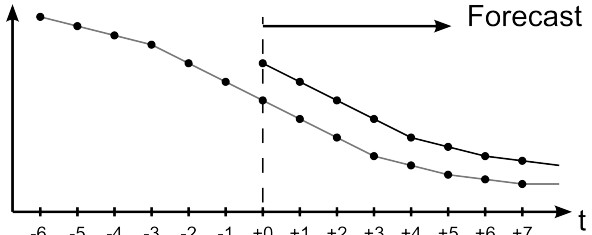

**Figure 1.** Conceptual time series of meteo data in a particular point for two consecutive forecasts: new (black line) and old (grey). Dots denote times, where data fields are available. The actual variable could be e.g. pressure, temperature or wind. The dashed line denotes the start of the meteorological forecast: +0H, *i.e.* the analysis time.

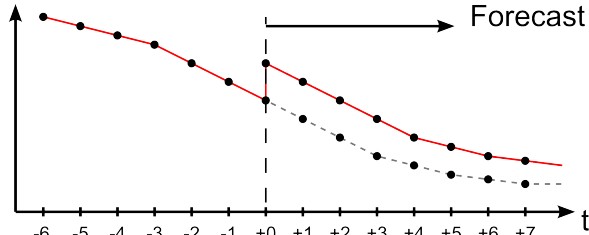

**Figure 2.** Conceptual example of meteorological forcing for ocean model with hotstart coinciding with the epoch. It is assumed that the ocean model uses linear interpolation between the available meteorological fields.

the initial fields of a forecast coincides in time with fields from several previous forecasts. The meteorological update cycle of hourly fields provided every six hours is used in illustrations and discussions in the present work. Even so, the results of the present work should be applicable to other update cycles as well.

In Figure 1, it is illustrated how a meteorological forecast differs from the previous forecast at a specific point. The difference is caused by the afforementioned data assimilation procedure in the meteorological forecasting system which adjusts the forecasts according to observations, and updates *e.g.* the location of low pressure areas. In the present work, the change from previous to new forcast at $t = 0$ will be denoted "the meteorological discontinuity" or simply "the jump". Some atmospheric models are adjusted within a short time window while others (*e.g.* 4DVAR) provide a smooth transition from the previous forecast to the current forecast in an assimilation window prior to the analysis time/epoch. It is however common practice to only disseminate forecasts from the analysis time and forward in time as illustrated in the figure. The dots shown on Figure 1 thus represents the data actually available for downstream use *e.g.* as forcing in oceanographic forecasting models.

## 2.2 Operational oceanographic forecasting

Operational ocean models are forced with meteorological forecast data received at each epoch. The ocean models typically run when the meteorological forecasts are available. However, the method for providing initial conditions to the subsequent forecast is rarely documented. It does however seem common to keep so-called "hotstart files" with the model state exactly at the analysis time, and then at time +6H save new hotstart files needed for the next epoch. At a first glance this choise of hotstart time may seem natural. However, as discussed, the meteorological data from two consecutive forecasts are discontinuous exactly at the epoch – the analysis time. Therefore, even though there is a high correlation between the current and previous forecasts, each meteo-

rological forecast should be seen as an entirely new forcing function for the ocean model rather than an extension of the previous forcing function, see Figure 2.

If the ocean model hotstart time is chosen to coincide with the analysis time, then effectively the ocean model is forced by meteorological fields from the previous forecast (gray line in Figure 1) up until the epoch and by new/updated fields after the epoch (black line in Figure 1). Even though the ocean model may interpolate (linearly) in time between the available meteorological fields, the forcing is discontinuous exactly at the epoch. This discontinuity in the meteorological forcing, which could include instantaneous repositioning of pressure minima, may generate non-physical waves, as the ocean model adapts to the sudden changes in forcing. Since this process is repeated every six hours (every epoch), the ocean model is in effect forced by discontinuous data every six hours back in time, and these discontinuities may generate spurious non-physical waves that propagate around the model domain.

It can be seen from Figure 2 that time -1H is the last meteo field, which is not changed by the new forecast. Thus, to avoid the discontinuity, the ocean model should really employ temporal interpolation from -1H (using data from the previous forecast) to +0H of the new forecast. However, as the new data for +0H is not available at the time of the previous ocean forecast, the new ocean forecast must start at -1H, see Figure 3. As a consequence, the hotstart files for the next forecast must be written at +5H; six hours of simulated time into the forecast. This approach corresponds exactly to running a hindcast based on the first six (hourly) fields of each meteo forecast, which – for hindcasts – seems to be a very common choice among ocean modellers. Modellers running hindcast according to this model (Figure 3) and operational/forecasts according to the previous (Figure 2) should be aware of the difference. In such cases, the hindcasts could under-predict the errors in the model leading to too high confidence in an operational model, when compared to the model in a hindcast scenario.

At the Joint GeoMETOC Support Center (FCOO), the ocean forecast always starts at -1H to avoid discontinuities

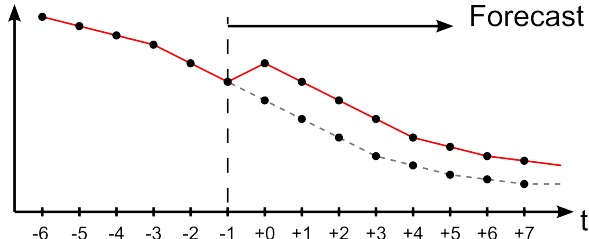

**Figure 3.** Conceptual example of meteorological forcing for ocean model with hotstart at -1H.

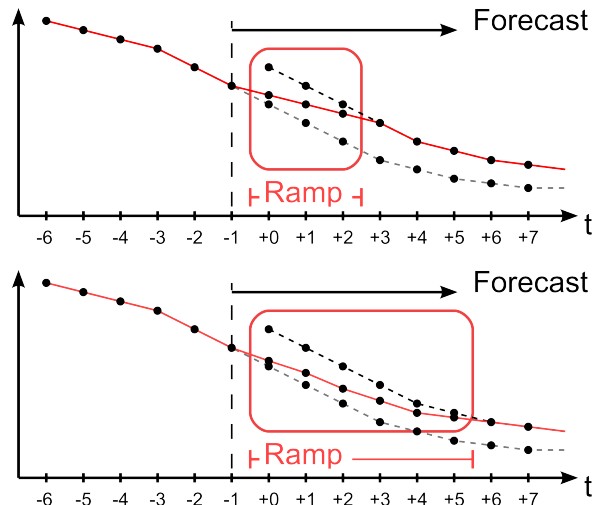

**Figure 4.** Conceptual examples of meteorological forcing for ocean model with hotstart at -1H and three or six hours of meteorological ramping. Longer ramping yields smoother transition between old and new forecast data.

of the meteorological forcing. The extra simulated hour is only a small overhead to pay for the increased model accuracy, which follows. Also, for hindcast simulations, FCOO use the same meteorological forcing principles as the opera-

5 tional model (excepting the forecast period itself), such that the hindcast performance may be used as a solid base for operational performance of at least the first 5-6 hours of the forecast.

### 2.3 Ramping

Even if hotstarts are positioned at -1H, it is not ensured that the meteorological forcing is smooth, see Figure 3. Although the severe discontinuity is avoided, the epoch can still represent an hour with larger changes of meteorology than a typical hour of the simulation. As a result there may still be

generated unphysical waves in the ocean model – although hopefully at a smaller scale. To increase the uniformity of the meteorological forcing, it is possible to perform a so-called ramping process, in which a linear combination between old and new meteorological forecast fields are used to change

smoothly from the old to the new forecasts. If $M_n^i$ denotes the $n$'th field of forecast no. i, so that $M_{n+6}^{i-1}$ and $M_n^i$ are consecutive forecasts for meteorological fields at the same time (hour on the clock), then the rampN process computes updated fields as

$$25 \quad \widetilde{M}_n^i = \frac{N-n}{N+1} M_{n+6}^{i-1} + \frac{n+1}{N+1} M_n^i \quad , \text{ for } n = 0, \ldots, N-1 \quad (1)$$

For $N > 6$, *i.e.* if the ramping is longer then the number of time fields between consecutive forecasts, then it is necessary to make the computation recursive, so that Eq. 1 is used for the first ramping ($i = 1$), while later epochs later epochs

use $\widetilde{M}_{n+6}^{i-1}$ in place of $M_{n+6}^{i-1}$. The method is outlined in Figure 4 for different values of the ramping length. It should be noted that the suggested ramping process does not imply interpolation in time.

The obvious disadvantage is that the ramping process sac-

35 rifices the accuracy of the first fields of the meteorological forecast, by replacing a significant part of the data with older and probably less accurate data from the previous forecast. Thus, the ramping is a delicate balance between smoothness

and (local) accuracy. As is the case in many other aspects of (ocean) modelling, such as the bathymetry, the use of the 40 most accurate (unfiltered) data does not implicitly ensure the most accurate results.

While the ramping processing of meteorological forcing data requires some pre-processing overhead, there is no computational penalty involved for the actual model forecasting. 45 It should be noted that it is entirely possible to use a ramp longer than the time between epochs.

Note that the initial shifting of the hotstart time shown in Figure 3 may be considered as a special case of the ramping procedure. If "rampN" denotes ramping over N meteo 50 fields, such that Figure 4 shows ramp3 and ramp6, then Figure 3 shows ramp0, i.e. a linear interpolation between old and new forecasts, but with no intermediate interpolated fields. The discontinuous process shown in Figure 2 could then be dubbed "no ramping" (noramp). 55

It should be noted, that if two consecutive meteorological forecasts provide very similar results, *i.e.* if the discontinuity is small, then the effect of ramping is also small. In essence, the suggested ramping process is not harmful in general, but "kicks in", when the discontinuity is significant. 60

At the Joint GeoMETOC Support Center (FCOO) all meteorological forcing data is ramped before used as forcing for oceanographic models. The ramping length depends on the experience with the particular meteorological dataset. For the ECMWF-IFS data, ramp6 is used operationally at FCOO. 65

If the weather centers would provide updated information for previous fields ($t < 0$) to present a smooth update from some prior known state to the present analysis time, then that would be preferable to using the presented ramping method. In such a scenario, the hotstart time of the oceanographic 70

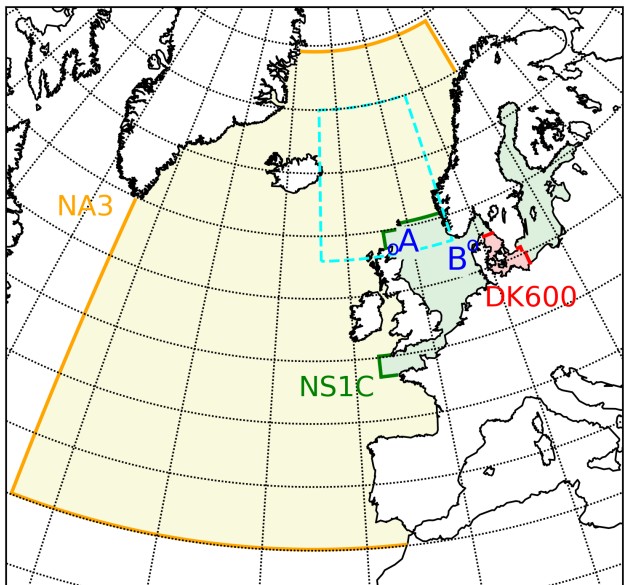

**Figure 5.** FCOO nested GETM ocean model setup: NA3, NS1C, and DK600. Station Wick (A) is used as proxy for effect on boundary conditions for the NS1C model, while station Hanstholm (B) is used to gauge the actual effect on the NS1C model. For later reference the north-east atlantic area "neatl" (lon=[-14:6] lat=[58:70]) is marked with a cyan dashed box.

model should simply be pushed back in time to the latest unchanged field, in a procedure totally equivalent to what is presented in Figure 3.

## 2.4 Operational ocean model setup

During each FCOO forecast cycle, three nested setups (Büchmann et al., 2011) of the General Estuarine Transport Model (GETM, see Burchard and Bolding, 2002) are run in sequence, see Figure 5. The outer-most setup (NA3) is a barotropic storm-surge model setup, and it feeds inwards to two nested 3D baroclinic setups (NS1C, DK600).

The NA3 setup is a 2D storm surge model with no tidal wave components included. The lateral boundary conditions are forced by Flather-style boundary conditions (Carter and Merrifield, 2007) driven by inverse barometric effect (Ponte et al., 1991) derived from the meteorological forecast from the ECMWF-IFS (Integrated Forecasting System) model. The meteorological wind (stress) and pressure components of the ECMWF-IFS forecast data also act directly on the ocean free surface.

The NS1C and DK600 setups are 3D baroclinic setups and include tides, which are added to the boundaries of NS1C, where OSU tidal data inversion (Egbert and Erofeeva, 2002) data are combined with surface elevations and depth-integrated velocities from the NA3 setup to get elevation and velocity boundary conditions. Data from DMI-HARMONIE-NEA (Bengtsson et al., 2017) are used to force the surface

of the NS1C and DK600 setups. As the NA3 setup is used for boundary conditions (nesting), there is an impact from the ECMWF-IFS model on the "inner" NS1C and DK600 models. Any discontinuities in the meteorological pressure – especially in the "neatl" area shown in Figure 5 – may propagate to the boundaries of the NS1C setup, and thus create non-physical waves in the NS1C and DK600 setups.[1]

Presently, FCOO makes 54 hour forecasts, i.e. to time +54H, of the entire model complex (NA3-NS1C-DK600) four times per day at epochs 00, 06, 12 and 18 UTC. The timing is based on data availability of the meteorological models used for forcing the ocean models.

## 3 Experiments and analysis

In this section, the meteorological pressure discontinuity will be examined in detail (§3.1). Subsequently, the effect on the ocean model will be examined (§3.2).

### 3.1 Meteorological forcing

In order to quantify the difference in sea level pressure (SLP) caused by the discontinuity between two meteorological forecasts, the typical SLP change during one hour of a forecast is computed, see Figure 6. For each of the four daily IFS-forecasts in 2016–2018, the absolute pressure difference between each forecast field and the previous field is computed for each of the first 18 fields (+01H to +18H). Subsequently, the values are averaged over each forecast (18 fields) and over all 4384 forecasts in the three-year period. For the "neatl" area near the northern boundary of the NS1C model (see Figure 5), the average hourly change is computed to 42 Pa.

The temporal average of the absolute value of the pressure jump has been computed, see Figure 7. For the open ocean, the magnitude is significantly lower than the average hourly pressure change shown in Figure 6: for the "neatl" area, the average "pressure jump" is computed to 0.22 hPa, or 54% of the average hourly change in the same area. In words, the update to the analysis field corresponds – on average – to about half the typical hourly change. However, the average pressure-change of Fig. 7 hides a rather large variability in both time and space. To examine the temporal variation, the spatial average (over the "neatl" area) of the absolute pressure jump has been computed as a function of time for the three-year period 2016–2018, see Figure 8. As written previously, the temporal average is 0.22 hPa, but this covers a variation exceeding an order of magnitude: from 0.068 hPa to 0.81 hPa, the latter occuring at 2016-12-24 00:00.

If the SLP discontinuity at 2016-12-24 00:00 is examined in detail, see Figure 9, then it may be seen that the local (in

---

[1]As NS1C and DK600 use DMI-HARMONIE-NEA data as forcing, the meteorological discontinuity of this model must be considered as well.

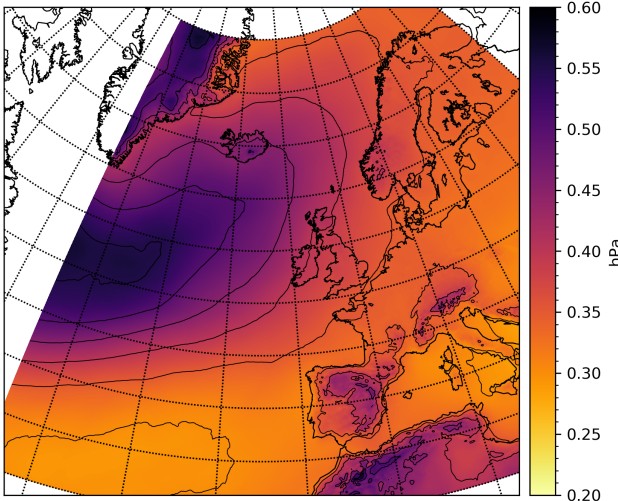

**Figure 6.** ECMWF-IFS absolute hourly pressure-change for the first 18 hours (+0H through +18H) forecast - averaged over all 4384 (four daily) forecasts in 2016–2018. The spatial average over the "neatl" area (Fig 5) is 0.42 hPa.

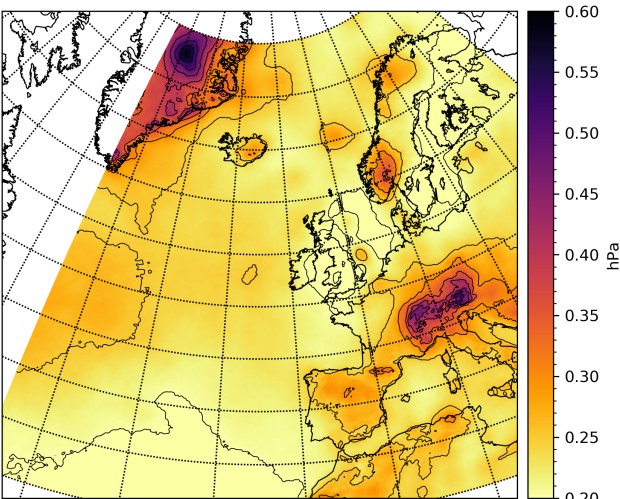

**Figure 7.** ECMWF-IFS absolute pressure disontinuity (jump) averaged over all forecasts in 2016–2018. The spatial average over the "neatl" area (Fig 5) is 0.22 hPa.

space) pressure-jump at that time exceeds 5 hPa. The largest pressure-jump appears on the east side of a low-pressure system (results not shown), where the meteorology analysis step apparently has adjusted the size and speed of the system slightly. Overall, however, the two pressure-fields (weather maps) do look vary similar, so there has not been any large error in the pressure-forecast.

Time series of SLP for three consecutive forecasts in a single point are depicted in Figure 10. For comparison, the time series for "noramp" and "ramp6" are also shown. It should be noted that without ramping, the SLP forcing used for a hydrodynamic model does include significant pressure

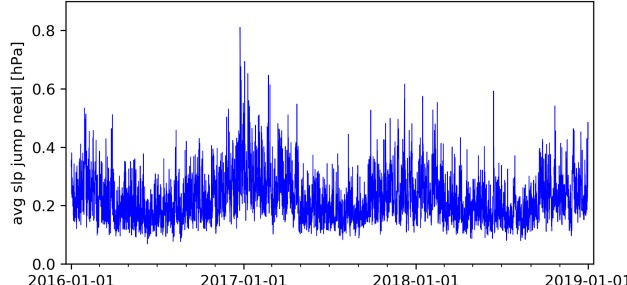

**Figure 8.** Spatial average over the "neatl" area (Fig 5) of ECMWF-IFS absolute pressure-jump for each forecast in 2016–2018.

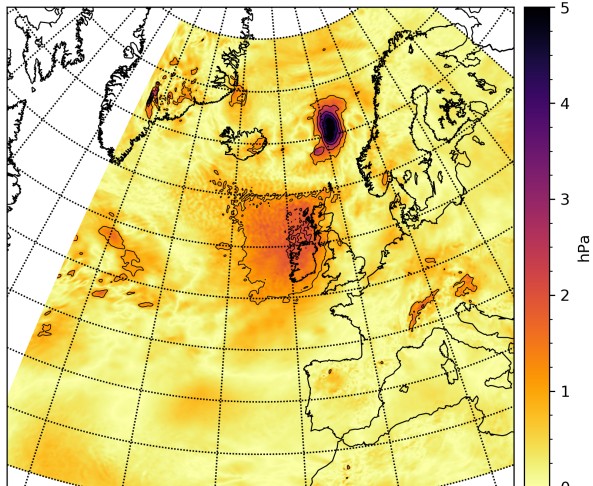

**Figure 9.** ECMWF-IFS absolute pressure-jump at 2016-12-24 00:00, ie absolute pressure difference between 2016122400+0H (analysis step) and 2016122318+6H. The spatial average over the "neatl" area (Fig 5) is 0.81 hPa. Note the changed color-scale: the maximum pressure-difference is 5.7 hPa, occurring at (lon,lat)=(0.773,64.793).

discontinuities. The introduction of ramping (see §2.3) de-

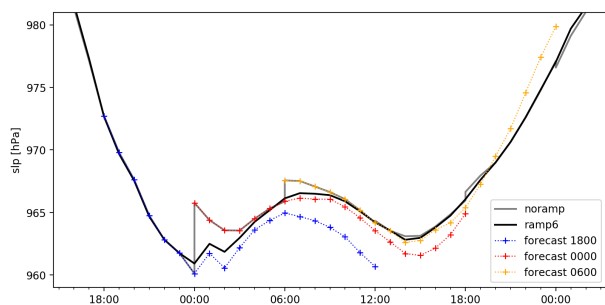

**Figure 10.** First 18 hours of ECMWF-IFS surface pressure (slp) forecasts at point (lon,lat)=(0.773,64.793), for epochs 2019-12-23 18:00, 2019-12-24 00:00, and 2019-12-24 06:00. Also shown are the resulting forcing without use of ramping ("noramp") and for a 6-hour ramping ("ramp6").

creases the spurious pressure jumps significantly. Further, if the pressure-jump is small, *ie.* if the present and previous forecasts agree, then there is no large impacts from the ramping procedure, and thus the noramp and ramp6 methods give essential the same result. Basically, the adverse effects of the ramping method seem to be small. Even though the impact of "best" first fields of each forecasts are reduced, the change is small unless there are large updates in the analysis step, which is exactly the time where ramping is necessary to alleviate discontinuities in the forcing.

### 3.2   Impact on ocean model

To investigate how the discontinuity affect the ocean elevation, the ocean models NA3 and NS1C have been run in hindcast mode with different preprocessing (ramping) of the meteorological forcing: noramp, ramp0 and ramp6, see figures 2, 3 and 4 for the period 2016-12-01 through 2018-12-31. In all cases, the ocean models have been forced by the ECMWF-IFS data set, only the ramping method is varied. In operational mode, the NS1C model is forced with meteo data from the Danish Meteorological Institute (DMI-HARMONIE-NEA), but for simplicity the ECMWF-IFS data is used in the present work.

As seen in the previous section, the pressure discontinuity may in certain situations exceed 5 $\mathrm{hPa}$. As a single hecto-Pascal corresponds roughly to one centimeter of water column, spurious waves in the order of several centimeters might be expected. As the discontinuous forcing could excite gravity waves over a range of frequencies, it may be difficult to *a priori* estimate the spectral content of the spurious waves.

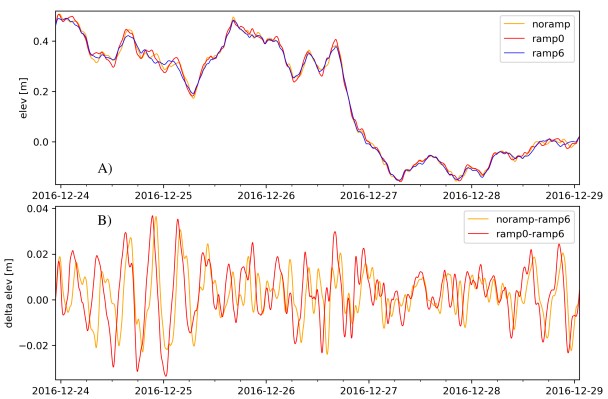

**Figure 11.** NA3 surge elevation at Wick station during December 2016 forced by ECMWF-IFS data. A: Results for models forced by the "noramp" (discontinuous), "ramp0", and "ramp6" methods, and B: differences between "short" ramps and the recommended "ramp6".

As a proxy for the impact on the NS1C north boundary, the computed surge elevation at Wick station (see Fig. 5) is

examined.[2] The time step ($\Delta t$) of the NA3 model is 8 $\mathrm{s}$, and to ensure that high-frequency components are resolved and to avoid data aliasing, station data has been saved every 40 $\mathrm{s}$, *ie.* every 5 time steps. In Figure 11, the Wick elevation is depicted for the two cases examined. Also the difference between the two surge time series is shown in the figure. With respect to the overall features – on the scale of several hours to days – the time series of the two methods agree. However, there is significant difference in the higher-frequency components: the elevation difference (Figure 11B) has significant energy components in the range of one to two hours (results not shown). For the chosen station and period, however, it is not clear if the "ramp0" method is significantly better than no ramping. It may be noted that the magnitude of the differences is in the order of several centimeters, consistent with the magnitudes of the pressure discontinuities found in the previous section. At FCOO, the modelled data (elevations, in this case) are routinely compared to observations in order to gauge the performance of the model. For the present setup and station, the RMS-error – comparing model elevation to observed surge (low-pass filtered elevation) – has been computed to 6.8 $\mathrm{cm}$ for 2016 and 5.1 $\mathrm{cm}$ for 2017. In comparison, the pressure discontinuity – inducing changes of up to 3–4 $\mathrm{cm}$ for the present station – is by no means the leading error term, but it is not negligible either.

The pressure discontinuity affects the nested model (NS1C) both directly on the free surface and through the boundary conditions computed from the NA3 results. For the NS1C setup, elevation data have been saved every 90 sec at more than 250 stations[3] throughout the domain. The Hanstholm station in the Western Skagerrak on the Danish peninsula of Jutland (see Fig. 5) is examined to gauge the effect on a station in the NS1C model setup. While the event in Dec 2016 results in differences of up to 4–5 $\mathrm{cm}$ at the Hanstholm station (results not shown, but data available online), a year later, the differences exceed 6 $\mathrm{cm}$, corresponding to nearly 5% of the monthly elevation range, see Figure 12. From the figure, it may be noted that the "noramp" method results in high-frequency components in the elevation signal, especially on 2017-12-26. In the present case, the "ramp0" method suppresses the major part of the high-frequency components. It is also observed that most of the time, the difference between the three methods is small. Compared to observed elevation, the statistics for the Hanstholm station (forced with ECWMF-IFS ramp6) shows an RMS-error of 6.9 $\mathrm{cm}$ for 2016 and 6.4 $\mathrm{cm}$ for 2017. Again, the pressure discontinuity is not a leading error term, but it should be worth to address.

For completeness, it should be mentioned that even the slightest change in forcing or initial conditions may induce a change the location of baroclinic eddies in the 3D circu-

---

[2]Time series of 40 stations for each of the three ramp settings are available online

[3]Time series for 266 stations are available online

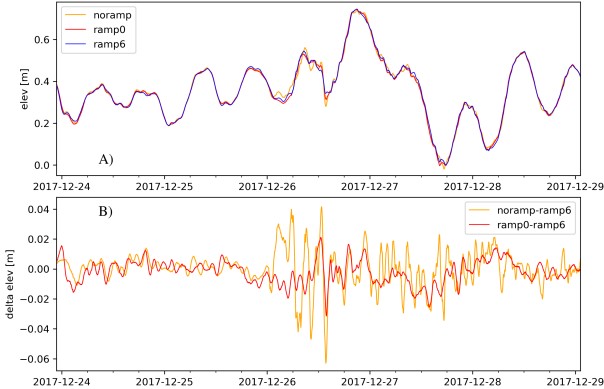

**Figure 12.** NS1C elevation at Hanstholm station during December 2017 forced by ECMWF-IFS data. A: Results for models forced by the "noramp", "ramp0" and "ramp6" methods, and B: differences between "short" ramps and the recommended "ramp6".

lation model. Büchmann and Söderkvist (2016) found the related salinity ensemble variability to be less than about 1PSU in Skagerrak. At any particular time the exact eddy location may differ between the three simulations (noramp, ramp0 and ramp6), and this introduces a small elevation difference, which is unrelated to the pressure jumps. The elevation difference was not examined explicitly for Skagerrak by Büchmann and Söderkvist (2016), but for a station in Kattegat the difference was found to be in the order of 0.1 mm – about two orders of magnitude smaller than the effects from the meteorological pressure discontinuities examined in the present work. Thus, the differences shown in Figure 12B, do originate from the pressure discontinuities rather than from repositioning of baroclinic eddies.

## 4   Conclusions

Data assimilation in meteorological models – implemented at the so-called "analysis step" – results in updated data fields at this and later time steps. It appears to be common practice, that weather centers, such as European Centre for Medium-range Weather Forecasts (ECMWF), disseminate updated forecast only from the forecast epoch ($t = 0$) and forward ($t > 0$). It has been shown that direct use of such meteorlogical forcing - in combination with saving hotstart files exactly at the epoch - results in the use of a meteorological forcing with a discontinuity at every model epoch back in time.

As a side-note it is observed that the discontinuity may not be present in hindcast mode, and thus the forecast-error of an oceanographical model could be under-estimated unless care is taken.

The size of the pressure discontinuity has been examined for a single model: the ECMWF-IFS model in a particular area in the North-East Atlantic. It was found that although the discontinuity on average is only about 0.2 hPa, it may ex-

ceed 5 hPa at specific incidents. A simple, straight-forward and easy-to-implement solution to the problem has been suggested, namely to slowly "ramp" the new forecast data, *ie.* to use a linear combination of old and new forecast for a few time steps.

The effect of the discontinuity on a particular operational storm-surge model has been examined, and spurious waves of the magnitude of several centimeters was observed in an area, where the storm-surge model data are used as boundary conditions for a higher-resolution operational circulation model. Additionlly, it has been shown that high-frequency spurious waves may also be seen directly in the circulation model. For the two particular stations examined, the magnitude of the spurious waves was in the same order of magnitude as the annual RMS-error of the model elevation compared to in-situ observations. Although the pressure discontinuity – for the present model setups – is not the leading error term, it is considered well worth the effort to eliminate the possible spurious waves in the models.

It is recommended that operational ocean modellers take steps to ensure that the meteorological forcing fields are smooth in time. Preferrably, meteorological forcing providers may provide updated forcing fields also before the forecast epoch, *ie.* for $t < 0$, thus to provide every field, which has been updated by the data assimilation procedure. In this case, the ocean model hotstart time should be moved to the time of the latest meteorological field, which is not updated by the process. As an alternative, the discontinuity may be alleviated by the ramping procedure suggested in the present work.

*Data availability.* Data used in this manuscript is available online at Zenodo.org, DOI:10.5281/zenodo.3243187. The raw data for each of the Figures 6–12 is available as indidual tar balls of NetCDF data files. Model elevation data are available in NetCDF data for 40 station in the NA3-domain and 266 stations in the NS1C-domain for the three examined ramp settings for the period 2016-12-01 through 2018-12-31.

ECMWF-IFS data are available through the MARS system, but access is limited to member countries. For the present paper, the first 19 hourly fields of each of four daily operational weather forecasts 2016-01-01 – 2018-12-31 have been used.

*Code availability.* GETM (Burchard and Bolding, 2002, https:// getm.eu/) is open source, and builds on GOTM (Umlauf and Burchard, 2005, https://gotm.net/) and FABM (Bruggeman and Bolding, 2014, https://github.com/fabm-model/fabm). The source code versions used in the present work reflects the executables in operational use at FCOO at the time of writing. As a consequence, the source codes differ between the NA3 and NS1C model setups. The exact git commit hashes and dates for the code used is given in the following table:

| Model | | branch | date | git commit |
|-------|------|------------|------------|-----------|
| NA3 | GETM | iow_master | 2017-04-21 | b23ad75 |
| | GOTM | iow_master | 2017-04-30 | 7fe7b19 |
| | FABM | master | 2016-12-01 | d5c7fbe |
| NS1C | GETM | iow_master | 2017-06-07 | 25dabf7* |
| | GOTM | iow_master | 2017-05-30 | a3fa955 |
| | FABM | master | 2016-12-01 | d5c7fbe |

For operational purposes the GETM code for NS1C was modified slightly to improve the initializations of particular arrays. The modifications does not impact on the results of the present paper. The exact source code – including changes – used in this study is available on-line at DOI:10.5281/zenodo.3243187. The actual changes compared to GETM-25dabf7 are given also as a diff-file.

*Author contributions.* BB identified the possible discontinuity, suggested the solution, examined the effects, implemented the solution operationally at FCOO, and wrote the paper.

*Competing interests.* The author declare that there is no conflict of interest

*Acknowledgements.* The author acknowledges fruitful discussions with and feedback from present and previous FCOO colleagues.

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
