# Peer review of "Dealing with discontinuous meteorological forcing in operational ocean modelling: a case study using ECMWF-IFS and GETM (v2.5)"

_Geoscientific Model Development, 2019_

## Referee Comment (RC1) · Anonymous Referee #1 · 16 Apr 2019

**General comments**

This paper concerns the use of operationally provided meteorlogical forcing for ocean models and the resulting discontinuities that arise between subsequent issued forecasts. This is an interesting topic and the paper is well written with clear explanations.

However, I feel that it needs more work to expand the results and put them in more context as we cannot currently tell whether the proposed ramping solution actually leads to any improvements in the ocean model forecasts.

[Figure]

**Specific comments**

The paper gives a convincing explanation of the hypothetical basis for introducing spurious waves into the ocean model from the discontinuous forcing, but does not give a feel for how much this is actually a problem in real forecasting. Has it caused noticeable issues with the operational surge model setup that is discussed?

The only comparison shown of results with and without the ramped forcing is at a single location (Wick) from the outer model NA3. The reasoning is that this is close to the boundary location of the nested NS1C model, but it would also be useful to see what differences are seen throughout the rest of the model domain to provide more information on the impact of the ramping method – is Wick representative of the rest of the region? Additionally it would be useful to see how the model results compare with observations at this location as we cannot currently tell if the ramping has led to an improvement. This would also put the differences between the "noramp" and "ramp6" cases in more context as I suspect the difference is small compared with other sources of model error.

Section 2.4 states that as the NA3 model is used to provide boundary conditions for the nested models any spurious waves created by the discontinuty in the NA3 model's forcing may therefore also enter the inner models through the boundaries. However there are no results shown or discussion of what (if any) differences are actually seen in the inner models when they are forced with boundary data from the two different experiments. Again it is therefore very difficult to judge the impact of the proposed ramping method.

It is acknowledged in the same section (in the footnote) that the inner models will have the same issue as their surface forcing dataset (DMI-HARMONIE-NEA) also contains the discontinuities - is there a reason the ramping experiment was only done for the outer model, where the potential impact is limited only to the effect on the boundary data that is used for the inner models? It would seem more relevant to try ramping the

forcing data of the inner models and see the impact that it has there as presumably they are the main focus of the operational system.

With these extra comparisons we would have a much clearer idea of what effect the ramping solution has on the ocean models. It may turn out to be the case that there is actually little impact, but that would still be a useful result since so many instutions run operational models forced by datasets containing these discontinuities.

**Technical corrections**

- The title! discontinuos -> discontinuous

- Page 7, line 5: apparantly -> apparently

- Page 10, line 14: have -> has

- Figures 6 & 7: It might be useful to include the location of the "neatl" box here as well?

- Figure 10: As a general point please avoid using red and green colours to distinguish lines, as this is not accessible to people with colourblindness. It would be better with a different pair of colours and/or different symbols on each line.

---

## Referee Comment (RC2) · Anonymous Referee #2 · 23 Apr 2019

General comment.

A priori, I was very interested in reviewing this paper because it addressed a question I had in mind since a long time and never found the time to analyze: the impact of the discontinuity introduced by the analysis step of the meteorological forcing used to drive operational ocean models and, more specifically, North Sea storm surge models. Unfortunately, the analysis is fairly limited: only a time series of surge elevation at station Wick without ramping and with ramp6 is presented and the difference between both simply indicate that spurious oscillations are generated when the no ramping method is applied. This is by far not enough to demonstrate the added value of the ramping method. Note also that this latter is, from my point of view, insufficiently explained. For instance, Figure 4 is supposed presenting the method for ramp3 and ramp6. If ramp3

seems really to be a linear interpolation between the field at t -1H and that at t + 4H this seems not to be the case for ramp6 (linear interpolation between t -1H and t + 6H). How is the method really working? But, the most disappointing point, from my point of view, is that no results are shown for the method the most commonly used in the operational oceanographic community according to the author himself, i.e., the ramp0 method. I would like to see this paper to be published but with a stronger demonstration of the benefit linked to the ramping method and, in particular with respect to the ramp0 approach.

Specific comments

As the paper needs a major revision, specific comments will be made on that version.

---

## Author Comment (AC1) · 25 Apr 2019

REPLY TO REVIEWER #1

If I read the comments of Reviewer #1 correctly, then the major general comment is to request additional results to gauge "whether the proposed ramping solution actually leads to any improvements in the ocean model forecasts". In particular the reviewer:

1. would like to see results for inner domains – with a query on why the experiment has not been made for these domains.

2. would like comparisons with measurements to see "if the ramping has led to an improvement".

I would very much like to respond to / explain each issue, and only subsequently suggest changes to the manuscript. I hope that the reviewer can then comment on the suggestions before I implement the changes to the manuscript.

Add 1:

We (FCOO) have been using meteo ramping for all our model setups (NA3, NS1C and DK600) for well over a decade. In that period, we have been through several "generations" of DMI-HIRLAM models (E15, T15, S05, S03), before recently (2018) switching to ECMWF-IFS for the outer domain and DMI-HARMONIE-NEA for the inner domains.

As it happens, the ECMWF-IFS model is rather well-behaved, presumably because the results are computed rather late in the 6-hour cycle, such that the first hours show only small change from previous forecast. I have worked with other meteo-datasets, where the problem was much more pronounced (in particular the now-retired DMI-HIRLAM-T15). Even so, ECMWF-IFS is a well-known and widely used model – presumably more so than data from a specific national weather centre, which is why I opted to disseminate this particular case.

For 3D (baroclinic) model setups (eg. NS1C, DK600), there is an additional complication, as even the slightest change of forcing or initial conditions, is likely to trigger a different outcome of the stochastic process, which is the model, see e.g. Büchmann and Söderkvist (2016) [DOI: 10.3402/tellusa.v68.30417]. In essence, the stochastic eddies might be positioned differently in two simulations, even if the changes to input are only at machine precision, and this may cause (small) changes also to the computed elevations. Fortunately, the effect on the elevations in the Kattegat area is rather small – sub-millimeter scale – so in principle it should be feasible to include modelled elevation for one or two positions from the inner domains: Kattegat and/or the south-western Baltic Sea.

Add 2:

It has not been my intention to show that ramping alone provides "better" results in term of, say, lowered RMS errors compared to elevation gauge measurements. That is why I have not included measurements – I find them irrelevant for the present discussion, and they could simply muddy the discussion (but see suggestions for changes later).

We routinely compute annual statistics for our models compared to measured elevations. For Wick, the RMSE error on surge (not tide) is typically in the order of 6-7 cm. The spurious waves stemming from the meteo discontinuity are smaller (several centimeters, see Fig 11B), so while they may not be the most important contribution to the model error, they should not be ignored either.

PLANNED REVISIONS

A:

I will make sure to note that FCOO operationally use ramping on all received meteo data before the data are used for modelling. I will also add a comment that the ramping length depends on our experience with the data set.

I will add a remark to the point that if the old and new meteo forecasts are very similar, ie. if the discontinuity is small, then the effect of ramping is also small (making a weighted mean of two similar results). In essence, the suggested ramping is not harmful, but "kicks in" in the cases, where the discontinuity is significant.

B:

I will include data (ocean model computed elevation) for one or two stations in shallower water/inner domain, presumably the Kattegat and/or the south-western Baltic Sea. These data will include effects of ramping (or not) of both the outer (ECMWF-IFS) and inner (DMI-HARMONIE-NEA) meteo models. Thus, the results will be less general, but maybe give a better sense for the scale of the problem. For DMI-HARMONIE-NEA we routinely use ramp9, and I will compare to noramp as well as to ramp0 (see reply to reviewer #2) data.

[Figure]

I plan to include the data in the same way as Fig 11, ie. time series of elevations and differences. I do, however, not plan to disseminate statistics on the alternative meteo model (DMI-HARMONIE-NEA).

C:

To give a better feel for whether this is "a problem in real forecasting", I will – for each station shown – include typical model errors (RMSE, compared to observed elevation), which can then be compared to the scale of the spurious waves.

COMMENTS ON TECHNICAL CORRECTIONS

1. The spelling corrections are duly noted. Thanks so much.

2. A typo in the title is just plainly embarrassing – sorry about that.

3. I have considered the possibility to add the "neatl-box" on figs 6, 7 and 9. Presently, I tend to think that the additional information "ready at hand" is not worth the increased complexity of the figures. In order to increase the readability, all figures already use exactly the same projection, although Fig 5 is slightly larger, as there is no colour legend. Unless the reviewer really would like to see this, I will not include the "neatl box" on additional plots.

4. I will try to find better colour scheme for the lines of Fig 10, replacing green with, say, orange.
* * *

---

## Author Comment (AC2) · 25 Apr 2019

**REPLY TO REVIEWER #2**

If I read the comments of Reviewer #2 correctly, then there are the following concerns:

1. The limited analysis, which the reviewer finds is "not enough to demonstrate the added value"

2. No results are given for "ramp0"

3. There is doubt about how the method is actually applied.
I would very much like to respond to / explain each issue and subsequently suggest changes to the manuscript. I hope that the reviewer can then comment on the suggestions before I implement the changes to the manuscript.

Add 1:

I have tried to keep the manuscript brief – in the spirit of creating a technical note. It was considered to include more data, but the decision was taken that more data might not necessarily make the manuscript more usable. Adding data for inner domains, such data would (still/also) be selective: the actual choice of meteo and ocean models would be specific to our present setup, and the gauge positions and periods would to some extend be arbitrary. Data from the "inner domains" (NS1C, DK600, see fig 5) might show a stronger response. However, that would introduce one or two extra ocean models as well as at least one extra meteo model, as another forcing (DMI-HARMONIE-NEA) is used for the inner domains.

We (FCOO) routinely compute annual statistics for our models compared to measured elevations. For Wick, the RMSE error on surge (not tide) is typically in the order of 6-7 cm. The spurious waves stemming from the meteo discontinuity are smaller (several centimeters, see Fig 11B), so while they may not be the most important contribution to the model error, they should not be ignored either.

The present choice of data was taken for the following reasons:

1. ECMWF-IFS is a well-known and widely used model – presumably more so than data from a specific national weather centre.

2. The actual size of the discontinuity varies with the meteo model/setup (as well as with time and space) and – at least to some extent – on the ocean model specifics, i.e. the magnitude of the problem will differ from case to case. Therefore, the choice was made to focus on the principles rather than specific values of the data.

As it happens, the ECMWF-IFS model is rather well-behaved, presumably because the results are computed rather late in the cycle, such that the first hours show only small change from previous forecast. I have worked with other meteo-datasets, where the problem was much more pronounced.

3. It is the goal of the paper to show that "there is a problem", and suggest a simple method to deal with it. The actual choice of ramping length (N) should depend on the application (weather model, update frequency, ocean model etc). In my opinion, "noramp" should never be used, as there is zero computational overhead going from noramp to ramp0. And modellers should at least be aware that there could be a problem – and in many cases N>0 should be considered.

4. It has not been my intention to show that ramping alone provides "better" results in term of, say, lowered RMS errors compared to elevation gauge measurements, or show improvements on particular time series.

Add 2:

The reviewer writes that "no results are shown for the method the most commonly used in the operational oceanographic community according to the author himself, i.e., the ramp0 method".

Firstly, it is not my intent to claim that ramp0 is "the most commonly used [method]". It is my impression (personal communications), that the ramp0 method is much used in *hindcast*, as written on p3, l19-20: "This approach [ramp0] corresponds exactly to running a hindcast based on the first six (hourly) fields of each meteo forecast, which seems to be a very common choice among ocean modellers."
However, in operational/forecast mode, even ramp0 would require that the hotstart

files are not written at epoch t=0, but at an earlier time (say, t=-1H). And unfortunately, it seems that that t=0 is commonly used for the hotstart files, thus leading to "no-ramp". Thus, it is my best guess (claim) that "noramp" is rather common in operational /forecast mode. For newer operational/forecast setups, where data assimilation of the recent past is included, the hotstart time is presumably "well before epoch". In such cases, even operational /forecast will typically end with a "ramp0" scenario.

Add 3:

The reviewer writes that "Figure 4 is supposed presenting the method for ramp3 and ramp6. If ramp3 seems really to be a linear interpolation between the field at t-1H and that at t+4H this seems not to be the case for ramp6 (linear interpolation between t-1H and t+6H). How is the method really working?"

This seems to be a point of genuine misunderstanding of the method. Obviously, this falls back on the author/manuscript, which means that I need to improve the section describing the method itself (see below). Basically, there is no interpolation in time taking place. Rather, each meteo-field is recomputed as a "linear combination" (weighted mean) between old forecast and new forecast – each taken for the same time value (hour of the clock). For instance, ramp3 uses 1/4 new value and 3/4 old value (ie 25/75) at t=0H, 50/50 at t=+1H and 75/25 at t=+2H.

The illustration for ramp3 (Fig 4) appears linear only because the two (old and new) forecasts exhibit near-linear behaviour over these few time slices.

**SUGGESTED REVISIONS**

Based on the reviewer comments, I plan to make the following changes to the manuscript. If possible, I should like (short) reviewer input, in order to limit the total

**[GMDD](GMDD)**
review-work.

1. As the "ramp3" part of Fig 4 seems to be misleading in the present version, I suggest striking that particular subfigure, such that Fig 4 will depict only ramp6.

2. I will improve the explanation of the method using a simple equation, adding around page 4 line 10:

If $M_n^i$ denotes the $n$'th field of forecast no. i, so that $M_{n+6}^{i-1}$ and $M_n^i$ are consecutive forecasts for meteorological fields at the same time (hour on the clock), then the rampN process computes updated fields as

$$\widetilde{M}_n^i = \frac{N-n}{N+1}M_{n+6}^{i-1} + \frac{n+1}{N+1}M_n^i \ , \ \text{ for } n = 0, \ldots, N-1 \tag{1}$$

For $N > 6$, it is necessary to make the computation recursive, so that Eq. 1 is used for the first ramping ($i = 1$), while later epochs use $\widetilde{M}_{n+6}^{i-1}$ in place of $M_{n+6}^{i-1}$. It should be noted that the suggested ramping process does not imply interpolation in time.

3. As suggested by the reviewer I plan to include "ramp0" data. I suggest to simply include a third panel on Fig 11, depicting difference between ramp0 and ramp6 data. That should indicate also to what extend ramp0 "is enough" compared to ramp6. Also, I will remedy the sentence on p3, l19-20 to stress that the "common choice" also is for hindcast:

This approach [ramp0] corresponds exactly to running a hindcast based on the first six (hourly) fields of each meteo forecast, which – for hindcasts – seems to be a very common choice among ocean modellers.

4. I will include data (ocean model computed elevation) for one or two stations in shallower water/inner domain, presumably the Kattegat and/or the south-western Baltic Sea, see suggestions from reviewer #1. These data will include effects of ramping (or not) of both the outer (ECMWF-IFS) and inner (DMI-HARMONIE-NEA) meteo models. Thus, the results will be less general, but maybe give a better sense for the scale of the problem. For DMI-HARMONIE-NEA we routinely use ramp9, and I will compare to noramp and ramp0 data.

For all stations I will include the typical RMSE error from our annual statistics, such that a comparison can be made between the size of the spurious waves – and the typical model error on the station.

I do not plan to include observation data in the figures (time series).
* * *

---

## Referee Comment (RC3) · Anonymous Referee #2 · 28 Apr 2019

Reply to the authors comments

Add 1

I understand the wish of the author "to keep the manuscript brief in the spirit of creating a technical note". Nevertheless, I do assume that for promoting the use (and getting the right to do it) inside FCOO of the ramping method, the author must have produced a more detailed report. I don't ask for such a detailed report but for something in between the actual version of the paper and something which could convince me I should use the ramp method. To be fair, I should say we are using the ramp0 method for years and would like to know if it possible to do a better job. I understand that the "magnitude of the problem differ from case to case", however, we do hope that the tendencies remains valid. For the model results from the inner domains (North Sea, e.g.), I have

no problem if they come from the NA3 implementation even if those results are not used in operational mode. If possible, this will avoid the mixture of models/model forcing.

Add 2

You know why I have a particular interest for the ramp0 method.

Add 3

A good description of the method (as it is foreseen, see below) will help to understand Figure 4.

Suggested revisions

1. If the method is well described, there is no need to change Figure 4.

2. The explanation of the method is good.

3. Thanks to add ramp0 data. Results at one station in the North Sea will be greatly appreciated as well.

---

## Short Comment (SC1) · 3 May 2019

This comment identifies aspects of this manuscript which do not comply with relevant GMD policy. These should be remedied in any revised submission.

1. Title. As per https://www.geoscientific-model-development.net/about/manuscript_types.html, where a development and technical paper relates to a particular model, then the model name and version need to be in the title. This is the case for this manuscript so the title should be adjusted accordingly.

2. Data availability. Even though the ECMWF data employed is licence restricted, some of your readers will have the appropriate licences. It is therefore still necessary to precisely identify which ECMWF data was used in which of your runs, so that readers

will be able to identify what you did.

3. Code availability. Simply pointing to the GETM website does not reliably enable the users to identify exactly which code you used and what you ran. Instead, you need to persistently and publicly archive the exact version of the code you used, as well as any input files and pre- or post-processing scripts. Many GMD authors find https://zenodo.org a good platform for this (indeed, since GETM uses git, you can use the Zenodo-github integration to partly automate the process. See: https://guides. github.com/activities/citable-code/).

---

## Author Response (AR1)

**Comments, replies – sorted by subject – with refs to changes in manuscript**

The following pages provide "a detailed point-by-point response" in "a clear and easy to follow sequence" with "(1) comments from Referees, (2) author's response, (3) author's changes in manuscript". Comments and replies are - of course - available on the GMD web site.

The comments are followed by a latexdiff between the originally submitted and the revised manuscript, as per instructions. It should be noted, that the latexdiff does not show changes to figures and references – nor to data availability or code availability sections. Two figures (Fig 5 and Fig 11) have been updated, one figure (Fig 12) has been added, and a single ref (Büchmann and Söderkvist, 2016) has been added – all in response to referee comments. Also the data availability and code availability sections have been significantly updated.

**Request for data from "inner domain" / additional station data**

**Referee 1:**

This paper concerns the use of operationally provided meteorological forcing for ocean models and the resulting discontinuities that arise between subsequent issued forecasts.

This is an interesting topic and the paper is well written with clear explanations.

However, I feel that it needs more work to expand the results and put them in more context as we cannot currently tell whether the proposed ramping solution actually leads to any improvements in the ocean model forecasts.

The paper gives a convincing explanation of the hypothetical basis for introducing spurious waves into the ocean model from the discontinuous forcing, but does not give a feel for how much this is actually a problem in real forecasting.

Has it caused noticeable issues with the operational surge model setup that is discussed?

The only comparison shown of results with and without the ramped forcing is at a single location (Wick) from the outer model NA3. The reasoning is that this is close to the boundary location of the nested NS1C model, but it would also be useful to see what differences are seen throughout the rest of the model domain to provide more information on the impact of the ramping method – is Wick representative of the rest of the region?

Section 2.4 states that as the NA3 model is used to provide boundary conditions for the nested models any spurious waves created by the discontinuty in the NA3 model's forcing may therefore also enter the inner models through the boundaries. However there are no results shown or discussion of what (if any) differences are actually seen in the inner models when they are forced with boundary data from the two different experiments. Again it is therefore very difficult to judge the impact of the proposed ramping method.

It is acknowledged in the same section (in the footnote) that the inner models will have the same issue as their surface forcing dataset (DMI-HARMONIE-NEA) also contains the discontinuities - is there a reason the ramping experiment was only done for the outer model, where the potential impact is limited only to the effect on the boundary data that is used for the inner models? It would seem more relevant to try ramping the forcing data of the inner models and see the impact that it has there as presumably they are the main focus of the operational system.

With these extra comparisons we would have a much clearer idea of what effect the ramping solution has on the ocean models. It may turn out to be the case that there is actually little impact, but that would still be a useful result since so many institutions run operational models forced by datasets containing these discontinuities.

**Author's reply to Referee 1:**

We (FCOO) have been using meteo ramping for all our model setups (NA3, NS1C and DK600) for well over a decade. In that period, we have been through several "generations" of DMI-HIRLAM models (E15, T15, S05,

S03), before recently (2018) switching to ECMWF-IFS for the outer domain and DMI-HARMONIE-NEA for the inner domains.

As it happens, the ECMWF-IFS model is rather well-behaved, presumably because the results are computed rather late in the 6-hour cycle, such that the first hours show only small change from previous forecast. I have worked with other meteo-datasets, where the problem was much more pronounced (in particular the now- retired DMI-HIRLAM-T15). Even so, ECMWF-IFS is a well-known and widely used model – presumably more so than data from a specific national weather centre, which is why I opted to disseminate this particular case.

For 3D (baroclinic) model setups (eg. NS1C, DK600), there is an additional complication, as even the slightest change of forcing or initial conditions, is likely to trigger a different outcome of the stochastic process, which is the model, see e.g. Büchmann and Söderkvist (2016) [DOI: 10.3402/tellusa.v68.30417]. In essence, the stochastic eddies might be positioned differently in two simulations, even if the changes to input are only at machine precision, and this may cause (small) changes also to the computed elevations. Fortunately, the effect on the elevations in the Kattegat area is rather small – sub-millimeter scale – so in principle it should be feasible to include modelled elevation for one or two positions from the inner domains: Kattegat and/or the south-western Baltic Sea.

I will make sure to note that FCOO operationally use ramping on all received meteo data before the data are used for modelling. I will also add a comment that the ramping length depends on our experience with the data set.

I will add a remark to the point that if the old and new meteo forecasts are very similar, ie. if the discontinuity is small, then the effect of ramping is also small (making a weighted mean of two similar results). In essence, the suggested ramping is not harmful, but "kicks in" in the cases, where the discontinuity is significant.

I will include data (ocean model computed elevation) for one or two stations in shallower water/inner domain, presumably the Kattegat and/or the south-western Baltic Sea. These data will include effects of ramping (or not) of both the outer (ECMWF-IFS) and inner (DMI-HARMONIE-NEA) meteo models. Thus, the results will be less general, but maybe give a better sense for the scale of the problem. For DMI-HARMONIE-

NEA we routinely use ramp9, and I will compare to noramp as well as to ramp0 (see reply to reviewer #2)

data.

I plan to include the data in the same way as Fig 11, ie. time series of elevations and differences. I do, however, not plan to disseminate statistics on the alternative meteo model (DMI-HARMONIE-NEA).

**Referee 2:**

A priori, I was very interested in reviewing this paper because it addressed a question I had in mind since a long time and never found the time to analyze: the impact of the discontinuity introduced by the analysis step of the meteorological forcing used to drive operational ocean models and, more specifically, North Sea storm surge models. Unfortunately, the analysis is fairly limited: only a time series of surge elevation at station Wick without ramping and with ramp6 is presented and the difference between both simply indicate that spurious oscillations are generated when the no ramping method is applied. This is by far not enough to demonstrate the added value of the ramping method.

**Author's reply to Referee 2:**

I have tried to keep the manuscript brief – in the spirit of creating a technical note. It was considered to include more data, but the decision was taken that more data might not necessarily make the manuscript more usable. Adding data for inner domains, such data would (still/also) be selective: the actual choice of meteo and ocean models would be specific to our present setup, and the gauge positions and periods would to some extend be arbitrary. Data from the "inner domains" (NS1C, DK600, see fig 5) might show a stronger response. However, that would introduce one or two extra ocean models as well as at least one extra meteo model, as another forcing (DMI-HARMONIE-NEA) is used for the inner domains.

The present choice of data was taken for the following reasons:

1. ECMWF-IFS is a well-known and widely used model – presumably more so than data from a specific national weather centre.
2. The actual size of the discontinuity varies with the meteo model/setup (as well as with time and space) and – at least to some extent – on the ocean model specifics, i.e. the magnitude of the problem will differ from case to case. Therefore, the choice was made to focus on the principles rather than specific values of the data.
   As it happens, the ECMWF-IFS model is rather well-behaved, presumably because the results are computed rather late in the cycle, such that the first hours show only small change from previous forecast. I have worked with other meteo datasets, where the problem was much more pronounced.
3. It is the goal of the paper to show that "there is a problem", and suggest a simple method to deal with it. The actual choice of ramping length (N) should depend on the application (weather model, update frequency, ocean model etc). In my opinion, "noramp" should never be used, as there is zero computational overhead going from noramp to ramp0. And modellers should at least be aware that there could be a problem – and in many cases N>0 should be considered.

I will include data (ocean model computed elevation) for one or two stations in shallower water/inner domain, presumably the Kattegat and/or the south-western Baltic Sea, see suggestions from reviewer #1. These data will include effects of ramping (or not) of both the outer (ECMWF-IFS) and inner (DMI-HARMONIE-NEA) meteo models. Thus, the results will be less general, but maybe give a better sense for the scale of the problem. For DMI-HARMONIE-NEA we routinely use ramp9, and I will compare to noramp and ramp0 data.

For all stations I will include the typical RMSE error from our annual statistics, such that a comparison can be made between the size of the spurious waves – and the typical model error on the station.

I do not plan to include observation data in the figures (time series).

**Referee 2:**

I understand the wish of the author "to keep the manuscript brief in the spirit of creating a technical note". Nevertheless, I do assume that for promoting the use (and getting the right to do it) inside FCOO of the ramping method, the author must have produced a more detailed report. I don't ask for such a detailed report but for something in between the actual version of the paper and something which could convince me I should use the ramp method.

I understand that the "magnitude of the problem differ from case to case", however, we do hope that the tendencies remains valid. For the model results from the inner domains (North Sea, e.g.), I have no problem if they come from the NA3 implementation even if those results are not used in operational mode. If possible, this will avoid the mixture of models/model forcing.

Results at one station in the North Sea will be greatly appreciated as well.

**Author's reply to Referee 2 response:**

Unfortunately, there is no recent detailed report, from which I can draw results.

I do not have a problem extracting results from one or two stations, more "central" for the FCOO area-of-interest. However, as NA3 is a simple storm-surge model, tidal data is simply not included. Presenting data from NA3 within the North Sea / Skagerrak area could be misleading, and I would rather not do that. This said, I shall be happy to produce the relevant data for either NS1C or DK600 – including tidal data. I realize, that this will mean mixing models/model forcing, but with a few lines of explanation, I believe that it should be OK. This should give some relevant order-of-magnitude, for differences between respectively noramp and ramp0 – compared to ramp9, which we use operationally for these models.

I will examine a few stations, but expect that data for the inner Danish waters should serve the purpose. I will wait a little longer for further comments from reviewer #1, and then I will prepare an updated version of the manuscript taking into account comments from both reviewers.

**Changes to manuscript:**

In the end it was decided to use ECMWF-IFS to force the "inner model" NS1C, even though another meteo forcing is used operationally. This decision was made to keep the manuscript as simple and clear as possible – while still accepting the comments and recommendations from the referees.

- Added paragraph at end of §2.3 that at FCOO all meteo data are ramped before use as forcing data
- Data from Hanstholm station added (Fig 12 + updates to §3.2 text)
- Added paragraph on ensemble variability in baroclinic models and reference to Büchmann and Söderkvist (2016).
- Update to Conclusions (§4)
- §3.2: Footnotes that GETM station data are available online

**1   Requests for comparison with observations**

**Referee 1:**

Additionally it would be useful to see how the model results compare with observations at this location as we
cannot currently tell if the ramping has led to an improvement. This would also put the differences between
the "noramp" and "ramp6" cases in more context as I suspect the difference is small compared with other
sources of model error.

**Author reply to referee 1:**

It has not been my intention to show that ramping alone provides "better" results in term of, say, lowered
RMS errors compared to elevation gauge measurements. That is why I have not included measurements – I
find them irrelevant for the present discussion, and they could simply muddy the discussion (but see
suggestions for changes later).

We routinely compute annual statistics for our models compared to measured elevations. For Wick, the RMSE
error on surge (not tide) is typically in the order of 6-7 cm. The spurious waves stemming from the meteo
discontinuity are smaller (several centimeters, see Fig 11B), so while they may not be the most important
contribution to the model error, they should not be ignored either.

**Referee 2:** (comment repeated from previous section)

Unfortunately, the analysis is fairly limited: only a time series of surge elevation at station Wick without
ramping and with ramp6 is presented and the difference between both simply indicate that spurious
oscillations are generated when the no ramping method is applied. This is by far not enough to demonstrate
the added value of the ramping method.

**Author reply to referee 2:**

We (FCOO) routinely compute annual statistics for our models compared to measured elevations. For Wick,
the RMSE error on surge (not tide) is typically in the order of 6-7 cm. The spurious waves stemming from the
meteo discontinuity are smaller (several centimeters, see Fig 11B), so while they may not be the most
important contribution to the model error, they should not be ignored either.

It has not been my intention to show that ramping alone provides "better" results in term of, say, lowered
RMS errors compared to elevation gauge measurements, or show improvements on particular time series.

**Changes to manuscript:**

- §3.2: values given for RMS-errors (model data compared to observations) for stations used
- §3.2+§4: RMS-errors compared to magnitude of the spurious waves induced by pressure jump, noting
that the pressure discontinuity may not be a leading error term, but that it is still worth to pursue.

**No results are given for "ramp0"**

**Referee 2:**

But, the most disappointing point, from my point of view, is that no results are shown for the method the most commonly used in the operational oceanographic community according to the author himself, i.e., the ramp0 method. I would like to see this paper to be published but with a stronger demonstration of the benefit linked to the ramping method and, in particular with respect to the ramp0 approach.

**Author reply to referee 2:**

The reviewer writes that "no results are shown for the method the most commonly used in the operational oceanographic community according to the author himself, i.e., the ramp0 method".

Firstly, it is not my intent to claim that ramp0 is "the most commonly used [method]". It is my impression (personal communications), that the ramp0 method is much used in hindcast, as written on p3, l19-20: "This approach [ramp0] corresponds exactly to running a hindcast based on the first six (hourly) fields of each meteo forecast, which seems to be a very common choice among ocean modellers."

However, in operational/forecast mode, even ramp0 would require that the hotstart files are not written at epoch t=0, but at an earlier time (say, t=-1H). And unfortunately, it seems that that t=0 is commonly used for the hotstart files, thus leading to "no-ramp". Thus, it is my best guess (claim) that "noramp" is rather common in operational/forecast mode. For newer operational/forecast setups, where data assimilation of the recent past is included, the hotstart time is presumably "well before epoch". In such cases, even operational /forecast will typically end with a "ramp0" scenario.

As suggested by the reviewer I plan to include "ramp0" data. I suggest to simply include a third panel on Fig 11, depicting difference between ramp0 and ramp6 data. That should indicate also to what extend ramp0 "is enough" compared to ramp6. Also, I will remedy the sentence on p3, l19-20 to stress that the "common choice" also is for hindcast:
This approach [ramp0] corresponds exactly to running a hindcast based on the first six (hourly) fields of each meteo forecast, which – for hindcasts – seems to be a very common choice among ocean modellers.

**Referee 2 response:**

To be fair, I should say we are using the ramp0 method for years and would like to know if it possible to do a better job.
You know why I have a particular interest for the ramp0 method.
3. Thanks to add ramp0 data.

**Author reply to referee 2 response:**

However, I shall be happy to produce the required data for ramp0. It is not a problem, except that it takes a little time – in particular the pre-processing of meteo data simply takes some wall time.

**Changes in manuscript:**

- Added "for hindcasts" in §2.2
- Added station Hanstholm (B) on Fig 5
- Fig 11 updated with "ramp0-ramp6" difference
- Some changes in text to include "ramp0" data, such as refs to figures etc.
- Comments on use of ramp0 vs noramp in §3.2

**1  How is the method actually working**

**2  Referee 2:**

Note also that this latter [the ramping method] is, from my point of view, insufficiently explained. For instance, Figure 4 is supposed presenting the method for ramp3 and ramp6. If ramp3 seems really to be a linear interpolation between the field at t -1H and that at t + 4H this seems not to be the case for ramp6

(linear interpolation between t -1H and t + 6H). How is the method really working?

**7  Author reply to Referee 2:**

The reviewer writes that "Figure 4 is supposed presenting the method for ramp3 and ramp6. If ramp3 seems really to be a linear interpolation between the field at t-1H and that at t+4H this seems not to be the case for ramp6 (linear interpolation between t-1H and t+6H). How is the method really working?"

This seems to be a point of genuine misunderstanding of the method. Obviously, this falls back on the author/manuscript, which means that I need to improve the section describing the method itself (see below).

Basically, there is no interpolation in time taking place. Rather, each meteo-field is recomputed as a "linear combination" (weighted mean) between old forecast and new forecast – each taken for the same time value (hour of the clock). For instance, ramp3 uses 1/4 new value and 3/4 old value (ie 25/75) at t=0H, 50/50 at t=+1H and 75/25 at t=+2H.

The illustration for ramp3 (Fig 4) appears linear only because the two (old and new) forecasts exhibit near- linear behaviour over these few time slices.

Author Suggested changes:

1.  As the "ramp3" part of Fig 4 seems to be misleading in the present version, I suggest striking that
particular subfigure, such that Fig 4 will depict only ramp6.
2.  I will improve the explanation of the method using a simple equation, adding around page 4 line 10:

<LATEX>

If $M^i_n$ denotes the $n$'th field of forecast no. i, so that $M^{i-1}_{n+6}$ and $M^i_n$ are consecutive forecasts for meteorological fields at the same time (hour on the clock), then the rampN process computes updated fields as

\begin{equation}

\widetilde{M}^{i}_n = \frac{N-n}{N+1} M^{i-1}_{n+6} + \frac{n+1}{N+1} M^i_n \enspace,\enspace \mbox{for\ }

n=0, \ldots , N-1

\end{equation}

For $N>6$, it is necessary to make the computation recursive, so that Eq.\ 1 is used for the first ramping ($i=1$), while later epochs use $\widetilde{M}^{i-1}_{n+6}$ in place of $M^{i-1}_{n+6}$.

It should be noted that the suggested ramping process does not imply interpolation in time.

</LATEX>

**35  Referee 2 response:**

A good description of the method (as it is foreseen, see below) will help to understand Figure 4.

1.  If the method is well described, there is no need to change Figure 4.
2.  The explanation of the method is good.

**39  Changes to manuscript:**

Update to §2.3 with explicit description on how method is applied – incl. Eq 1.

Added paragraph at end of §2.3 that method is not harmful in general, but "kicks in" when the discontinuity is significant.

**Figures 6+7**

**Referee 1:**

Figures 6 & 7: It might be useful to include the location of the "neatl" box here as well?

**Author reply to Referee 1:**

I have considered the possibility to add the "neatl-box" on figs 6, 7 and 9. Presently, I tend to think that the additional information "ready at hand" is not worth the increased complexity of the figures. In order to increase the readability, all figures already use exactly the same projection, although Fig 5 is slightly larger, as there is no colour legend. Unless the reviewer really would like to see this, I will not include the "neatl box" on additional plots.

**Changes to manuscript:**

No further comments have been received on this subject.

No changes made to Fig 6+7.

**Figure 10**

**Referee 1:**

Figure 10: As a general point please avoid using red and green colours to distinguish lines, as this is not accessible to people with colourblindness. It would be better with a different pair of colours and/or different symbols on each line.

**Author reply to Referee 1:**

I will try to find better colour scheme for the lines of Fig 10, replacing green with, say, orange.

**Changes to manuscript:**

Fig 10 updated – both changing green to orange, and use of different symbols/markers on the coloured lines.

**Comments on spelling**

**Referee 1:**

1. The title! discontinuos -> discontinuous
2. Page 7, line 5: apparantly -> apparently
3. Page 10, line 14: have -> has

**Author reply to Referee 1:**

1. The spelling corrections are duly noted. Thanks so much.
2. A typo in the title is just plainly embarrassing – sorry about that.

**Changes to manuscript:**

Typos corrected.

**Manuscript title**

**Editor comment:**

As per https://www.geoscientific-model-development.net/about/manuscript_types.html, where a development and technical paper relates to a particular model, then the model name and version need to be in the title. This is the case for this manuscript so the title should be adjusted accordingly.

**Author reply to editor:**

The reviewers have requested inclusion of data from more stations, and this may mean that I will also need to use meteo-data from DMI (Danish weather service) HIRLAM-NEA. These data are (unfortunately, and due to restrictions imposed by DMI) not publicly available, but I will of course describe what is going on. The final title will be updated with "a case study with ECMWF-IFS, DMI-HARMONIE-NEA and GETM (v2.5)".

**Changes to manuscript**

In the end it was decided to present results of the inner model forced by ECMWF-IFS. Title updated to include "a case study with ECMWF-IFS and GETM (v2.1)"

**Code availability**

**Editor comment:**

Simply pointing to the GETM website does not reliably enable the users to identify exactly which code you used and what you ran. Instead, you need to persistently and publicly archive the exact version of the code you used, as well as any input files and pre- or post-processing scripts. Many GMD authors find https://zenodo.org a good platform for this (indeed, since GETM uses git, you can use the Zenodo-github integration to partly automate the process. See: https://guides.github.com/activities/citable-code/).

**Author reply to editor:**

I will put the exact (GETM) source code (probably in a tarball) alongside all data I am at liberty to give on an open server, such that it can be evaluated along-side the revised manuscript. I find scripts to download, format and convert meteo-data well beyond scope. The actual method suggested to prepare (ramp) the meteo will be included in a revised paper, as per response to reviewer 2.

**Changes to manuscript:**

- Actual code - which differs slightly between the two operational setups - have been gathered and stored at Zenodo.org, and released with an DOI tag:
- Actual code tags - and link to online code repository (at zeonodo) - are now given in the manuscript code section.

**1 Data availability**

**Editor comment:**

Even though the ECMWF data employed is licence restricted, some of your readers will have the appropriate licences. It is therefore still necessary to precisely identify which ECMWF data was used in which of your runs, so that readers will be able to identify what you did.

**Author reply to editor:**

I am not sure exactly what is meant by "precisely identify which ECMWF data was used". Is this the actual field names, the horizontal resolution or something else?

The temporal resolution, areas of coverage and the periods are already explicitly mentioned. Running a (semi)operational model is so complex a task, that it is not possible for me to upload all data files, scripts and configurations in a way such that another user will be able to get exactly the same result - or even run them on a different system. I will strive to release as much data as possible - and in particular the raw data necessary for any figure used in the final paper. It is not clear, however, if data such as hotstart files for our particular GETM setups have any relevance for readers - and such files could actually contain data which we are not at liberty to release. This said, I am all in favour of open data, and I will strive to release as much as is possible. However, if GMD will insist on a release of ALL "input files and pre- or post-processing scripts", then I

cannot comply, and the paper thus cannot be released in GMD. If that is the case, then I should like to be informed as soon as possible, such that I may consider other publication options.

**Changes to manuscript:**

• Reference to data availability added to manuscript: DOI:10.5281/zenodo.3243187
• Updated information on ECMWF-IFS data sue and availability.
• Data for all stations from the two GETM setups released (at zenodo) for all ramping cases, in high
temporal resolution covering 25 months of simulation. Data are available as one NetCDF file per
station per month, and collected in one tarball (tgz-file) per ramping case.
• Processed data for all figures released on Zenodo

[revised manuscript text omitted]